# Right-Sided Minimally Invasive Direct Coronary Artery Bypass: Clinical Experience and Perspectives

**DOI:** 10.3390/medicina59050907

**Published:** 2023-05-09

**Authors:** Florian Hecker, Mascha von Zeppelin, Arnaud Van Linden, Jan-Erik Scholtz, Stephan Fichtlscherer, Jan Hlavicka, Thomas Walther, Tomas Holubec

**Affiliations:** 1Department of Cardiovascular Surgery, University Hospital Frankfurt and Goethe University Frankfurt, 60598 Frankfurt, Germany; florian.hecker@kgu.de (F.H.); mascha.vonzeppelin@kgu.de (M.v.Z.); arnaud.vanlinden@kgu.de (A.V.L.); jan.hlavicka@kgu.de (J.H.); thomas.walther@kgu.de (T.W.); 2Department of Radiology, University Hospital Frankfurt and Goethe University Frankfurt, 60388 Frankfurt, Germany; jan-erik.scholtz@kgu.de; 3Department of Cardiology, University Hospital Frankfurt and Goethe University Frankfurt, 60388 Frankfurt, Germany; stephan.fichtlscherer@kgu.de

**Keywords:** minimally invasive cardiac surgery, coronary artery bypass grafting (CABG), minimally invasive direct coronary bypass grafting (MIDCAB), anomalous right coronary artery (ARCA)

## Abstract

*Objective:* Minimally invasive direct coronary artery bypass grafting (MIDCAB) using the left internal thoracic artery to the left descending artery is a clinical routine in the treatment of coronary artery disease. Far less is known on right-sided MIDCAB (r-MIDCAB) using the right internal thoracic artery (RITA) to the right coronary artery (RCA). We aimed to present our experience in patients with complex coronary artery disease who underwent r-MIDCAB. *Materials and Methods:* Between October 2019 and January 2023, 11 patients received r-MIDCAB using RITA to RCA bypass in a minimally invasive technique via right anterior minithoracotomy without using a cardiopulmonary bypass. Underlying coronary disease was complex right coronary artery stenosis (*n* = 7) and anomalous right coronary artery (ARCA; *n* = 4). All procedure-related and outcome data were evaluated prospectively. *Results:* Successful minimally invasive revascularization was achieved in all patients (*n* = 11). There were no conversions to sternotomy and no re-explorations for bleeding. Furthermore, no myocardial infarction, no strokes, and, most importantly, no deaths were observed. During the follow-up period (median 24 months), all patients were alive and 90% were completely angina free. Two patients received a repeated revascularization after surgery but independently from the RITA-RCA bypass, which was fully competent in both patients. *Conclusion:* Right-sided MIDCAB can be performed safely and effectively in patients with expected technically challenging percutaneous coronary intervention of the RCA and in patients with ARCA. Mid-term results showed high freedom from angina in nearly all patients. Further studies with larger patient cohorts and more evidence are needed to provide the best revascularization strategy for patients suffering from isolated complex RCA stenosis and ARCA.

## 1. Introduction

Treatment of coronary artery disease (CAD) is being performed using interventional or surgical techniques. Percutaneous coronary intervention (PCI) has become rapidly advanced and broadly available, leading to guideline adoption and an overall reduction in surgical myocardial revascularization. Especially isolated single- and double-vessel CAD is predominantly treated by PCI. Nowadays, MIDCAB has become a well-established surgical routine treatment for proximal left anterior descending (LAD) stenosis, chronic totally occluded LAD, or in high-risk patients with multivessel CAD as an isolated treatment or as part of a hybrid revascularization strategy [1,2].

Characterized by a small skin incision and access (left anterior minithoracotomy), avoidance of sternotomy and no need for cardiopulmonary bypass could prove favorable, leading to significantly shorter ventilation time, less perioperative trauma, and thus a reduction in hospital length of stay [3,4].

While left-sided MIDCAB using the left internal thoracic artery (LITA) for grafting the LAD has been widely reported, there is less evidence about right-sided MIDCAB (r-MIDCAB) using the right internal thoracic artery (RITA) for grafting the right coronary artery (RCA). Isolated surgical revascularization of the RCA is very rare and, in most cases, treated by PCI. However, there are still some specific indications in complex coronary artery disease where the surgical approach is the only possible option.

The aim of the current study was to evaluate the feasibility and safety, as well as short- and mid-term outcomes of r-MIDCAB in patients with complex CAD of the RCA.

## 2. Materials and Methods

### 2.1. Study Design and Patients’ Characteristics

This single-center study involves 11 patients who underwent isolated r-MIDCAB procedures at our institution between October 2019 and January 2023. All patients received RITA to RCA bypass using a minimally invasive technique via right anterior minithoracotomy without using a cardiopulmonary bypass. Indications for surgery included the following: 1. complex CAD in seven patients; and 2. anomalous right coronary artery (ARCA) in four patients.

### 2.2. Patient Management and Preoperative Planning

All patients received a routine preoperative coronary angiography, transthoracic echocardiography (TTE), and standard chest X-ray to determine the right cardiac silhouette. In addition, patients received cardiac computed tomography (CT) to facilitate pre- and post-operative planning. 3Mensio Structural Heart, version 10.2. (SP1-Build 5233.47) (3Mensio Medical Imaging, Bilthoven; The Netherlands) was used for three-dimensional reconstruction of the CT scans for checking the feasibility of the minimally invasive approach and for optimal planning, including the position of the skin incision and minithoracotomy (Figure 1).

### 2.3. Surgical Technique

In all patients, unilateral lung ventilation, established by double-lumen tracheal intubation or by selective endoluminal blockage under bronchoscopic guidance, was performed. Transesophageal echocardiography was used in all patients. The patients were placed in the supine position on the table, and the chest was elevated 30° on the right side with a soft pillow underneath the scapula.

Right-sided MIDCAB was performed through a 5–6 cm long right anterior minithoracotomy, and RITA was harvested under direct vision using a Braun/Hauser retractor (Braun/Aesculap AG; Melsungen, Germany) (Figure 2). The RITA harvesting was performed using the standardized skeletonized technique for improved wound healing using a regular diathermal blade and hemostatic clips. In contrast to the left-sided MIDCAB, where the 4th intercostal space is entered for LITA harvesting, in the majority of the r-MIDCAB patients, the 5th intercostal space was used to reach the maximum length of the RITA (Figure 3) and better exposure of the RCA. Proximally, the RITA was harvested completely up to the 1st intercostal space to ligate all intercostal arteries and distally until the bifurcation. Systemic heparin (1 mg/kg) was then administered to achieve an activated clotting time of at least 300 s during the surgery. Then the RITA was divided, and the distal end was fixed to the skin level using a 6-0 polypropylene suture. The pericardium was opened ventrally over the right atrium at a length of approximately 7–9 cm, and pericardial stay sutures were placed for ventral luxation of the heart. A reusable mechanical MIDCAB stabilizer (Fehling Instruments; Karlstein, Germany) was used for stabilization of the planned target area of the anastomosis to the RCA. The RCA was proximally snared using a silastic hollow vessel loop with a blunt-tipped needle. The anastomosis was performed in a regular end-to-side off-pump fashion using 8-0 polypropylene sutures, routine use of a coronary shunt, and a CO_2_ blower for visual clearance of the operative field. After completion of the anastomosis, the bypass was checked by transit time flow measurement (TTFM) (MiraQ Cardiac; Medistim ASA, Oslo, Norway). The pericardium was closed and a 28F chest drain placed into the right pleural cavity; thereafter, the lung was reinflated under direct vision. The minithoracotomy was closed using a costal resorbable polyglyactin 2-0 Z-suture, followed by the layered wound closure and administration of periosteal and intramuscular 0.7% bupivacaine.

### 2.4. Postoperative Management

Most patients were extubated in the operating room after checking their neuromuscular state and transferred to the intensive care unit afterwards. Acetylsalicylic acid at a dose of 100 mg per day was started 1 day after surgery and is recommended for life. In addition, dual antiplatelet therapy with either Ticagrelor 90 mg twice daily or Clopidogrel 75 mg daily was started on the second postoperative day for 12 months after surgery to enhance the graft patency rate.

### 2.5. Data Collection, Follow-Up and Outcome

Pre-, peri-, and postoperative data were collected prospectively. Follow-up was performed in all patients by out-patient visits or phone calls, and additional information was provided by general physicians and cardiologists. Particularly, data on survival, freedom from angina, number of hospitalizations, and freedom from major cerebral and cardiac events were collected.

### 2.6. Statistical Analysis

Continuous and discrete variables were expressed as mean ± SD or median and interquartile range (IQR) for data not normally distributed. Categorical and ordinal variables were expressed by the number and percentage of observations. A statistical analysis was performed using Stat view (Carry, NC, USA) and IBM SPSS (version 25 for MS Windows; IBM Corporation, Armonk, NY, USA).

## 3. Results

### 3.1. Patients’ Demographics and Preoperative Data

Patients’ demographics and preoperative data are summarized in Table 1. The average age of the patients was 62.4 ± 15.3 years, and 4 patients were female. All patients were symptomatic with CCS class II–III, and the majority had prior myocardial infarction (55%), and prior PCI (64%). Referral for surgery was due to failed PCI (33%) or in-stent re-stenosis with emerging symptoms (66%). Most of the patients (89%) were elective; one patient received an urgent operation due to a failed PCI and a high risk of total occlusion. More than half of the patients were diagnosed with dyslipidemia (46%). Nearly all of our patients had at least kidney failure level I (90%).

### 3.2. Intra- and Early Postoperative Outcomes

The intra- and early postoperative outcomes are presented in Table 2. A successful r-MIDCAB was performed in all patients. There were no conversions to sternotomy. The mean total surgery time was 192.2 ± 48.6 min. There were no re-explorations for bleeding. Furthermore, no excessive bleeding, no myocardial infarction, no stroke, and no acute kidney injury were observed. All patients were discharged from the hospital alive, with a mean length of stay of 7.5 ± 3.2 days. The mean transient time flow measurements (TTFM) were 59.3 ± 43.2 mL/min in the CAD group and 62.8 ± 16.5 mL/min in the ARCA group. The pulsatility index (PI) was 1.7 ± 0.4 and 1.8 ± 0.3, respectively.

### 3.3. Follow-Up

The detailed follow-up data is presented in Table 3. The median follow-up was 24 months (range 0.7–42 months) and was 100% complete. All patients were alive, and ten patients (90%) were symptom-free at the follow-up. One patient (CAD group) received repeated revascularization with PCI of a progressing CAD-lesion distal to the RIMA anastomosis after 8 months. The RIMA-RCA bypass was fully patent. One patient in the ARCA group had a pathologic ECG finding 12 h post-surgery and was catheterized. RITA-RCA was patent with excellent run-off, but an acute in-stent thrombosis in the LAD was revealed, which was implanted 4 weeks prior to surgery and received acute PCI.

## 4. Discussion

The right-sided MIDCAB procedure is rarely described in the literature, and we present our rather large experience with this technique in patients with complex CAD. In analogy to the well-established MIDCAB technique with LIMA-LAD grafting, we believe r-MIDCAB is a very valuable technique for minimally invasive treatment of complex CAD of the RCA in selected patients. There were two main indications for the surgery (ARCA and CAD) in our cohort, and we also focused on the perspectives for both.

ARCA is a very rare coronary anomaly and has a prevalence of 0.23% in the general population [5]. The potential hemodynamically relevant interarterial course of the ARCA has been classified as a “malignant variant” with potential risk for sudden cardiac death [6]. The overall management of ARCA in adults is still under discussion, and surgical therapy is currently recommended only in symptomatic patients or in asymptomatic patients with diagnosed ARCA and reproducible ischemia on exercise testing due to a decreasing risk of sudden cardiac death with age [7]. Although mostly affecting young adults during high physical activity, ARCA may be represented by sudden cardiac death as a first symptom [6,8]. Research on anomalous coronary origins with or without reported sudden cardiac death could not highlight any significant differences in coronary anatomical landmarks in these patients. Only age over 30 was a significant factor correlated with a decreased risk of sudden cardiac death [7]. In our series, all three ARCA patients were clearly symptomatic, were in grades CCS II-III, and were therefore referred for surgical therapy. The main three surgical options for ARCA treatment are coronary artery bypass grafting, transposition of the coronary ostium to its appropriate sinus, and unroofing of the coronary artery in its intramural portion [9,10,11]. The latter two require cardio-pulmonary bypass or at least partial sternotomy, are therefore potentially more invasive, and may be prone to unexpected difficulties [10]. Although unroofing and transposition lead to an anatomical correction of ARCA, the long-term outcomes of both techniques remain unreported [12]. In the early years of MIDCAB, Izhar et al. published the first successful case report of r-MIDCAB in ARCA. Following this concept, Reddy et al. published their series of four patients receiving r-MIDCAB in ARCA later on [13,14]. Due to our experience and quite high and stable numbers (about 50 patients) of regular left-sided MIDCAB procedures every year, we share the opinion of Reddy et al. that the minimally invasive r-MIDCAB approach is the most straight-forward treatment approach for ARCA with its normal distal artery anatomy [14].

The data from our study support the safety and feasibility of the r-MIDCAB in ARCA. There was no conversion to sternotomy and no 30-day mortality in our cohort. One patient received acute repeated PCI due to early in-stent thrombosis of the LAD, which was implanted 4 weeks prior to surgery. The RITA-RCA bypass was fully patent and had perfect run-off. During the follow-up, no recurrent angina was observed in three out of four patients, and no further treatments were needed. One patient was re-hospitalized due to polymorphic symptoms and classified as grade CCS II. He underwent a cardiac CT scan followed by catheterization, showing a patent RITA-RCA bypass with excellent run-off (Figure 4). The first two patients who were treated with r-MIDCAB in ARCA were planned for re-catheterization half a year after surgery for quality control. In both cases, surgical results were excellent, with a fully patent graft and excellent run-off. Due to the dynamic character of the coronary stenosis in ARCA, r-MIDCAB may be prone to graft failure or even occlusion due to competitive flow in the native RCA [15]. Proximal native RCA ligation and the use of saphenous vein grafts were proposed to solve that problem [16,17]. Facing that problem, transient time flow measurements were performed in all patients and were precisely evaluated. The current evidence suggests that the minimum acceptable TTFM in CABG should be 15–20 mL/min and that the pulsatility index (PI) should be <3 [18,19]. With respect to one-lung ventilation, there is evidence that pulmonary vascular resistance increases due to hypoxia-induced vasoconstriction [20]. Due to the intramural and interarterial course of the RCA, the dynamic stenosis of the ARCA is directly influenced by aortic and pulmonary artery pressure. Taken this into consideration, we postulate that during r-MIDCAB in ARCA with normal blood pressure, there is an increased pulmonary artery pressure, which will lead to a consecutive increase in the grade of RCA stenosis, which may result in a falsely high TTFM flow and low PI, just as in higher physical activity. Therefore, we raised our cut-off values in TTFM in ARCA and expected good graft patency in flows >40 mL/min and PI < 2. One of the four patients had an initial TTFM of 6 mL/min with a PI of 9. Therefore, an RCA ligation was performed, resulting in a flow of 50 mL/min with a PI of 1.4. The other three patients had excellent TTFM, and no ligation was performed.

The regular CAD patients in this series were all symptomatic with persistent angina ranging in CCS grade II-III. The majority of the patients (86%) had prior PCI attempts. Interventional treatment was associated with either unsuccessful revascularization due to a demanding anatomy (43%) or led to in-stent re-stenosis (ISR; 57%) in a dominant RCA. As expected, all of the patients had a relatively low mean SYNTAX score of 8.9 ± 6.0 and were therefore not initially referred to surgical revascularization. The treatable lesions with the r-MIDCAB technique are in RCA segments one and two and, depending on anatomy, in proximal segment three (Figure 5). The patency rate of RITA graft has been shown to be excellent, with 89% at 10 years in angiographic observational studies, and was shown to stay excellent even at 15 years with a patency rate of 91% in a very recent study and is therefore comparable to LITA [21,22]. Although modern iterations of drug-eluting stents significantly reduced the incidence of ISR, there is still a high prevalence of ISR at about 5–10% of all performed PCIs [23,24]. One of the contributing factors to ISR is a lesion length of more than 20 mm and ostial lesions [25]. In the current revascularization guideline, there is no clear recommendation for a preferred treatment for isolated RCA stenosis with its relatively low SYNTAX score; moreover, isolated RCA stenosis may be preferably treated by PCI [26]. Additionally, LITA-LAD could provide favorable long-term data compared to PCI in multivessel disease in the BEST and the FREEDOM trials [27,28]. Furthermore, in the case of ISR in drug-eluting stents, the rates of major adverse cardiovascular events are shown to be higher than in de-novo stenosis, with up to 17% [29]. Current guidelines have a lack of evidence, so in some circumstances, surgical revascularization, even as a first-line recommendation, may be superior to PCI, especially in ostial or complex stenosis. Davierwala et al. presented their data on over 2667 MIDCAB patients and pointed out that the r-MIDCAB patients were excluded [1]. While r-MIDCAB is a technically challenging surgery, and currently most of those patients remain treated by PCI, it would be of quite some importance to evaluate isolated RITA-RCA bypass surgery against PCI to close the lack of evidence. Our current data could prove good mid-term results in r-MIDCAB with total freedom from angina. Although one patient (14.3%) needed re-revascularization of a downstream RCA vessel, it is unlikely that this is linked to the r-MIDCAB procedure; in addition, surgical revascularization is not known to raise the risk for PCI in the future. Repeat revascularization without further details in MIDCAB was reported by Davierwala et al. with 5.5% in 2667 patients in 22 years [1]. Furthermore, this study showed no reexploration for bleeding, no myocardial infarction, no acute kidney injury, and, moreover, no in-hospital mortality. Regarding those adverse events, Davierwala et al. reported 1.9% reexlorations for bleeding, 0.7% myocardial infarction, 1.0% acute kidney injury, 0.3% wound infections, and an in-hospital mortality of 0.9% in left-sided MIDCAB surgery [1].

### Study Strengths and Limitations

To the best of our knowledge, the current series is the largest cohort of r-MIDCAB patients that has been published yet.

This study has several limitations. It is based on a single-center experience with a new operative technique, the number of patients is very small and the follow-up is just mid-term.

## 5. Conclusions

This series provides evidence that r-MIDCAB is feasible and safe, with very good mid-term results. More evidence is needed to provide the best revascularization strategy to patients suffering from isolated RCA stenosis and ARCA; therefore, larger and possibly multicenter studies with prospective randomized trials are warranted.

## Figures and Tables

**Figure 1 medicina-59-00907-f001:**
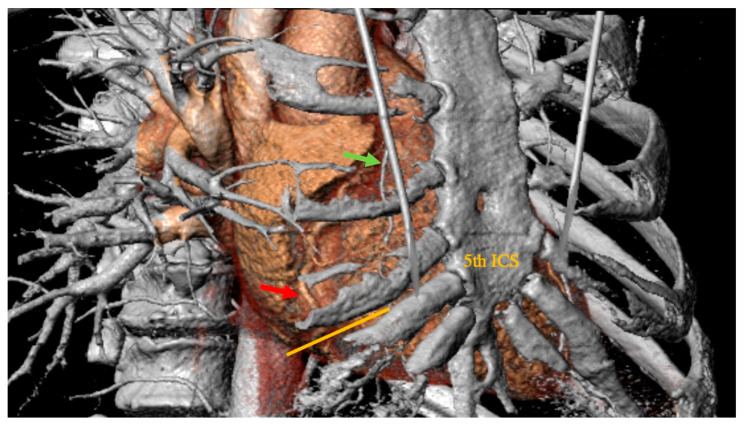
Preoperative thoracic computer tomographic scan evaluated in 3Mensio Structural Heart (3Mensio Medical Imaging; Bilthoven, The Netherlands), which is showing the right coronary artery (red arrow points to the planned site of anastomosis) in projection to the 5th intercostal space. Green arrow: right internal mammary artery. Orange line: planned skin incision.

**Figure 2 medicina-59-00907-f002:**
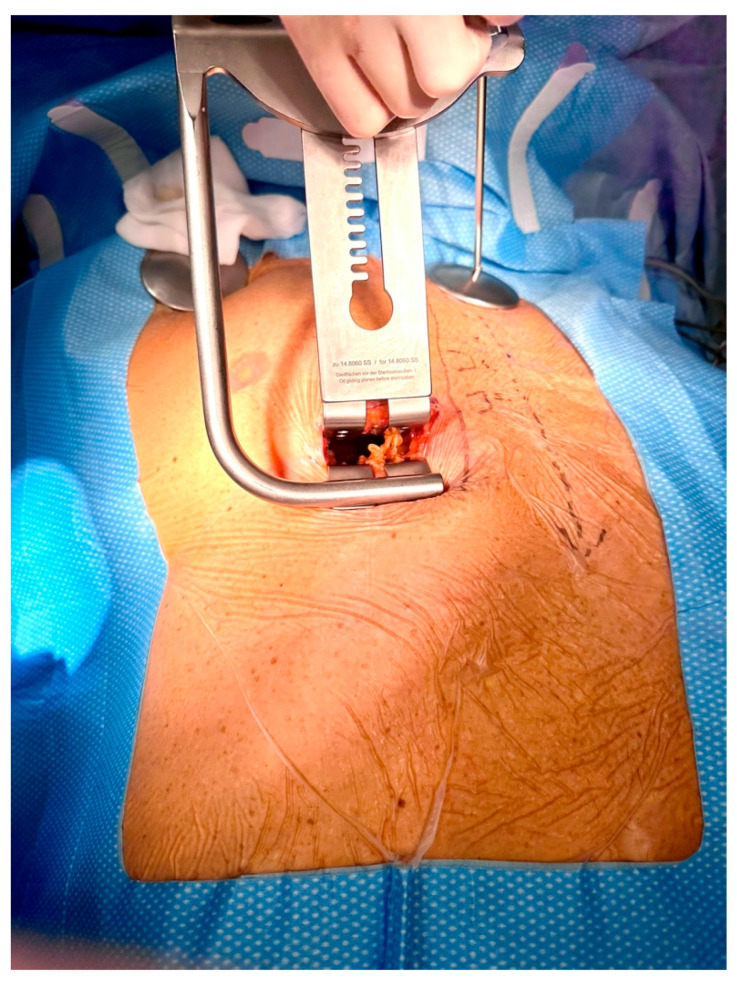
Intraoperative picture showing surgical site with inserted Braun/Hauser MIDCAB retractor (Braun/Aesculap AG; Melsungen, Germany) in the 5th intercostal space for preparation of the right internal thoracic artery.

**Figure 3 medicina-59-00907-f003:**
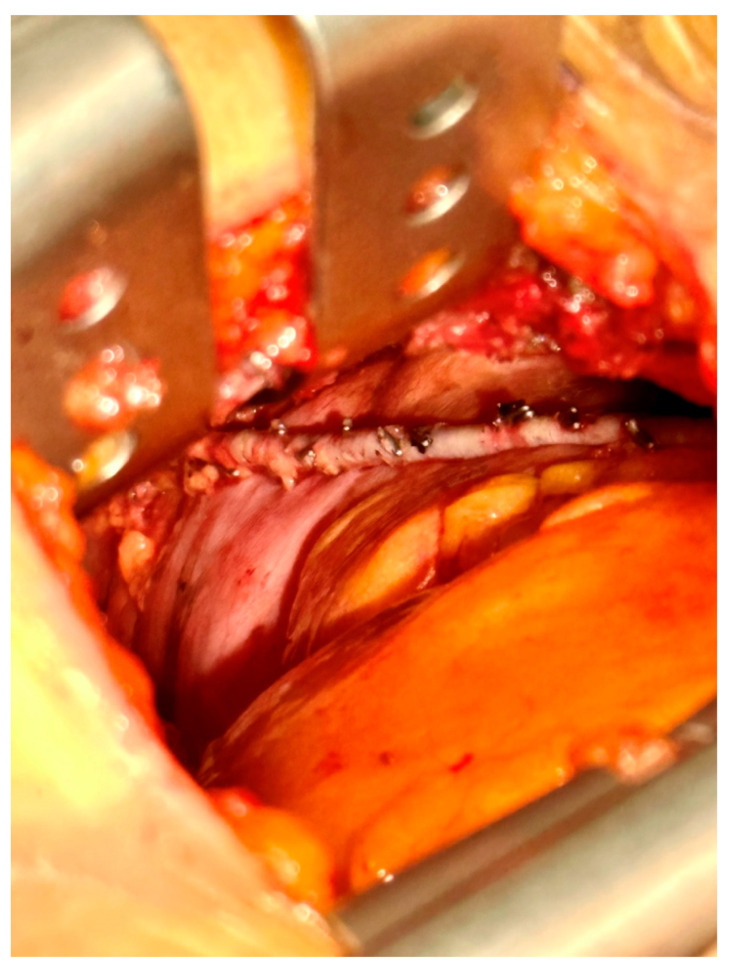
Intraoperative picture of the surgical site with surgical view showing the skeletonized right internal thoracic artery in situ.

**Figure 4 medicina-59-00907-f004:**
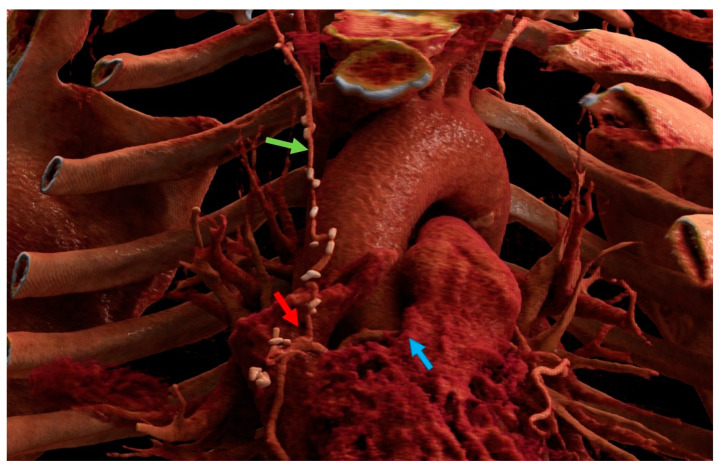
Cinematic rendering image of one of the anomalous right coronary artery patients after right-sided minimally invasive coronary artery bypass grafting. Blue arrow: intraarterial course of the right coronary artery between the aorta and pulmonary artery). Green arrow: right internal mammary artery. Red arrow: anastomosis. Images created with Cinematic Anatomy based on Cinematic Rendering (Siemens Healthineers; Erlangen, Germany).

**Figure 5 medicina-59-00907-f005:**
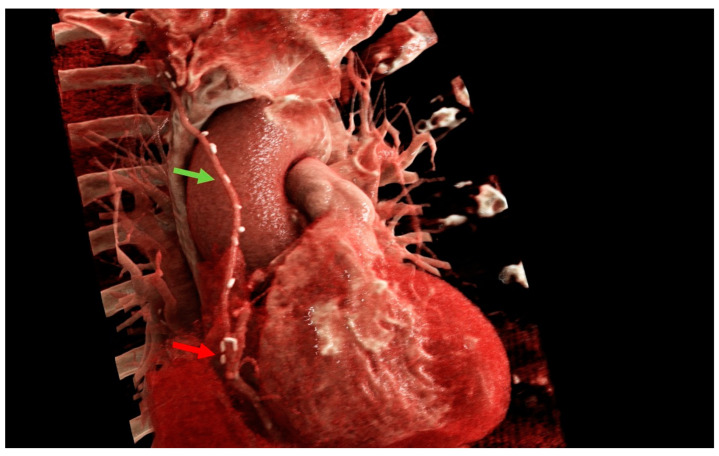
Cinematic rendering images of one of the coronary artery disease patients after a right-sided minimally invasive coronary artery bypass grafting procedure. Green arrow: right internal mammary artery. Red arrow: anastomosis. Images created with Cinematic Anatomy based on Cinematic Rendering (Siemens Healthineers; Erlangen, Germany).

**Table 1 medicina-59-00907-t001:** Patients’ preoperative data and comorbidities in the overall cohort and in the CAD and ARCA subgroups.

Variable		CAD (*n* = 7)	ARCA (*n* = 4)	Total (*n* = 11)
		Number/%/Mean ± SD	Number/%/Mean ± SD	Number/%/Mean ± SD
Age (years)		60.1 (±18.2)	66.3 (±8.9)	62.4 (±15.3)
Female		3 (42.9)	2 (50.0)	5 (45.5)
BMI		27.8 (±3.7)	35.4 (±5.2)	30.6 (±5.5)
Smoker		4 (57.1)	3 (75.0)	7 (63.6)
Diabetes		2 (28.6)	3 (75.0)	5 (45.5)
	insulin	1 (14.3)	1 (25.0)	2 (18.2)
Dyslipidemia		4 (57.1)	1 (25.0)	5 (45.5)
COPD		1 (14.3)	1 (25.0)	2 (18.2)
PAD		3 (42.9)	1 (25.0)	4 (36.4)
AF		1 (14.3)	0 (0)	1 (9.1)
Previous stroke		1 (14.3)	0 (0)	1 (9.1)
Kidney failure				10 (90.1)
	I	1 (14.3)	0 (0)	1 (9.1)
	II	5 (71.4)	3 (75)	8 (72.7)
	III	0 (0)	1 (25.0)	1 (9.1)
	IV	0 (0)	0 (0)	0 (0)
Ejection fraction		57.3 (±6.4)	62.5 (±5.0)	59.2 (±6.2)
Previous myocardial infarction		4 (57.1)	2 (50.0)	6 (54.5)
CCS				
	0	0 (0)	0 (0)	0 (0)
	1	2 (28.6)	0 (0)	2 (18.2)
	2	1 (14.3)	4 (100)	5 (45.5)
	3	3 (42.9)	0 (0)	3 (27.3)
	4	1 (14.3)	0 (0)	1 (9.1)
NYHA functional class				
	I	1 (14.3)	2 (50.0)	3 (27.3)
	II	3 (42.9)	2 (50.0)	5 (45.5)
	III	3 (42.9)	0 (0)	5 (45.5)
	IV	0 (0)	0 (0)	0 (0)
SYNTAX score				
Previous PCI		6 (85.7)	2 (50.0)	7 (63.6)
	LAD	3 (42.9)	2 (50.0)	5 (45.5)
	RCX	2 (28.6)	1 (25.0)	3 (27.3)
	RCA	5 (71.4)	0 (0)	5 (45.5)
Indication for surgery				
	ARCA	0 (0)	4 (100)	4 (36.4)
	In-stent re-stenosis	4 (57.1)	0 (0)	4 (36.4)
	failed PCI	3 (42.9)	0 (0)	3 (27.3)
Urgency				
	elective	6 (85.7)	4 (100)	10 (90.1)
	urgent	1 (14.3)	0 (0)	1 (9.1)
EuroScore II		1.7 (±0.6)	2.8 (±2.4)	2.1 (±1.5)

BMI: body mass index; COPD: chronic obstructive pulmonary disease; PAD: peripheral artery disease; AF: atrial fibrillation; CCS: Canadian Cardiovascular Society classification of angina; NYHA: New York Heart Association classification of dyspnea; PCI: percutaneous coronary intervention; LAD: left anterior descending artery; RCX: circumflex artery; RCA: right coronary artery; ARCA: anomalous right coronary artery; SD: standard deviation.

**Table 2 medicina-59-00907-t002:** Intra- and early postoperative results in the overall cohort and in the CAD and ARCA subgroups.

Variable		CAD (*n* = 7)	ARCA (*n* = 4)	Total (*n* = 11)
		Number/%/Mean ± SD	Number/%/Mean ± SD	Number/%/Mean ± SD
Total surgical time (minutes)		203.3 ± 58.5	172.8 ± 15.1	192.2 ± 48.6
Total length of stay (days)		7.3 ± 4.0	7.8 ± 1.5	7.5 ± 3.2
Ventilation time (minutes)		335.6 ± 97.4	385.5 ± 43.8	353.7 ± 83.1
Successful revascularisation		7 (100)	4 (100)	11 (100)
Wound infection		1 (14.3)	0 (0)	1 (9.1)
Tansient time flow measurement				
	mL/min	59.3 ± 43.2	62.8 ± 16.5	59.3 ± 34.7
	PI	1.7 ±0.4	1.8 ± 0.3	1.8 ± 0.3
In-hospital mortality		0 (0)	0 (0)	0 (0)

SD: standard deviation PI: pulsatility index.

**Table 3 medicina-59-00907-t003:** Patients´ follow-up data.

Variable		CAD (*n* = 7)	ARCA (*n* = 4)	Total (*n* = 11)
		Number/%/Mean ± SD	Number/%/Mean ± SD	Number/%/Mean ± SD
Angina (CCS)				
	0	7 (100)	3 (75)	10 (90.1)
	I	0 (0)	0 (0)	0 (0)
	II	0 (0)	1 (25)	1 (9.1)
	III	0 (0)	0 (0)	0 (0)
	IV	0 (0)	0 (0)	0 (0)
Revascularization after surgery		1 (14.3)	1 (25)	2 (18.2)
Re-hospitalization		1 (14.3)	0 (0)	1 (9.1)
Alive		7 (100)	4 (100)	11 (100)

SD: standard deviation.

## Data Availability

The data underlying this article will be shared on reasonable request to the corresponding author.

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
