# Peer review of "Right-Sided Minimally Invasive Direct Coronary Artery Bypass: Clinical Experience and Perspectives"

_medicina, 2023, doi:10.3390/medicina59050907_

Round 1

Reviewer 1 Report

The authors describe their good outcome of right-sided MIDCAB. I think this manuscript is well written and it is worthwhile to be published in this journal. One suggestion is to add a comment that the distal anastomosis of the RITA graft is proximal part of the RCA. So in the CAD group, the lesion should be in the ostium of the RCA. 

Author Response

Responses to reviewers´ comments (Manuscript ID: medicina-2336225)

Legend: 1) comment of reviewer 2) our response 3) changes

Reviewer #1:

Comment #1:

  1. One suggestion is to add a comment that the distal anastomosis of the RITA graft is proximal part of the RCA. So in the CAD group, the lesion should be in the ostium of the RCA.
  2. Thank you very much for your comment. We added a more detailed description into the main text. The most distal lesion reachable with a RITA graft in minimally invasive technique is segment 3, so the lesion for bypass in CAD group is able to treat in segments 1, 2 and in the proximal portion of segment 3.
  1. See page 9 line 257 to 259.

Reviewer 2 Report

This clinical study shows that right-sided minimally invasive direct coronary artery bypass grafting can be performed safely and effectively in patients with expected technically challenging percutaneous coronary intervention of the right coronary artery and in patients with anomalous right coronary artery. Hence, this study is of great significance for the therapy of coronary heart disease, and is very interesting to the cardiovascular surgeons.

Nevertheless, further studies with larger patient cohorts and more evidence are needed to provide the best revascularization strategy to the patients suffering from coronary heart disease.

Additionally, data related to the changes of inflammatory markers are also significant, as indicated by the following references, which may be discussed.

[1]     Urbanowicz T, OlasiÅ„ska-WiÅ›niewska A, Michalak M, Rodzki M, Witkowska A, StraburzyÅ„ska-Migaj E, Perek B, Jemielity M. The Prognostic Significance of Neutrophil to Lymphocyte Ratio (NLR), Monocyte to Lymphocyte Ratio (MLR) and Platelet to Lymphocyte Ratio (PLR) on Long-Term Survival in Off-Pump Coronary Artery Bypass Grafting (OPCAB) Procedures. Biology (Basel). 2021 Dec 27;11(1):34. doi: 10.3390/biology11010034.

[2]     Urbanowicz T, Michalak M, Al-Imam A, OlasiÅ„ska-WiÅ›niewska A, Rodzki M, Witkowska A, Haneya A, Buczkowski P, Perek B, Jemielity M. The Significance of Systemic Immune-Inflammatory Index for Mortality Prediction in Diabetic Patients Treated with Off-Pump Coronary Artery Bypass Surgery. Diagnostics (Basel). 2022 Mar 4;12(3):634. doi: 10.3390/diagnostics12030634.

[3]     Chen YW, Lee WC, Fang HY, Sun CK, Sheu JJ. Coronary Artery Bypass Graft Surgery Brings Better Benefits to Heart Failure Hospitalization for Patients with Severe Coronary Artery Disease and Reduced Ejection Fraction. Diagnostics (Basel). 2022 Sep 16;12(9):2233. doi: 10.3390/diagnostics12092233.

Author Response

Reviewer #2:

Comment #1:

  1. Nevertheless, further studies with larger patient cohorts and more evidence are needed to provide the best revascularization strategy to the patients suffering from coronary heart disease. Additionally, data related to the changes of inflammatory markers are also significant, as indicated by the following references, which may be discussed…
  2. Thank you very much for your inspiring comment! We read the two papers of Urbanowicz et al. and the study of Chen et al. with great interest. Basic/inflammatory science seems to be a very promising and important point in treatment planning in the future and we are very interested in looking into that. Maybe this could be an interesting follow-up research topic for another study. Due to the fact that we clearly focused on the surgical alternative strategy, this may sum up to a too broad spectrum for our currently presented study.
  3. No changes were done because we decided that the inflammatory response may be presented and analyzed separately.

Reviewer 3 Report

Hecker et alt provided an interesting experience in applying minimally invasive coronary artery bypass. The work is generally well-written, and in particular, I appreciated the description of the methodology and the discussion. 

However, some issues remain.

How did you perform patient selection? All ischemic patients failed PCI?

11 patients are few. How did you estimate it was an adequate number for your observational prospective study? How much is the overall number of CABG in your center?

What is the expected rate of adverse peri-procedural and 24-months events? Substantially, you didn't have adverse events; however, the sample size could be under-powered, and some relevant complications could be messed

Can you report the type and number of adverse events in the discussion with literature data, even regarding left MIDCAB?

Author Response

Reviewer #3:

Comment #1:

  1. How did you perform patient selection? All ischemic patients failed PCI
  2. Thank you very much for your comment! The presented study was more or less an all-comers cohort. We actually included all patients of our center who were presented with isolated RCA stenosis and evaluated all of them for r-MIDCAB due to our relatively high experience in regular MIDCAB surgery. Failed PCI was the reason in all of our CAD patients, yes.
  3. No changes were done.

Comment #2:

  1. 11 patients are few. How did you estimate it was an adequate number for your observational prospective study? How much is the overall number of CABG in your center?
  2. Thank you very much for your valuable comment. Indeed, 11 patients is very low number. However, this will be one of the largest cohorts published ever. And further, we clearly stated this a feasibilty and safety study and want to present our data as an inspiration for other centers and research groups to publish their data on r-MIDCAB. It would be very interesting to get more information from other experience MIDCAB centers. Our overall number on regular CABG surgery is about 500 patients a year with about 30% in off-pump manner. Minimally-invasive CABG with regular MIDCAB (LIMA-LAD) is stable with about 60 patients per year.
  3. No changes were made.

Comment #3:

  1. What is the expected rate of adverse peri-procedural and 24-months events? Substantially, you didn't have adverse events; however, the sample size could be under-powered, and some relevant complications could be messed.
  2. Thank you for your comment. The expected rate of adverse peri-procedural events and the 24-months events would probably be comparable to the presented data of Davierwala et al (DOI: 10.1016/j.jtcvs.2020.12.149) who provided a 22 years experience in over 2000 patients receiving MIDCAB surgery. Our provided sample size may be under-powered but is nevertheless very interesting for inspirational reasons to other groups to evaluate their results in this fairly rare surgical method.
  3. No changes were made.

Comment #4:

  1. Can you report the type and number of adverse events in the discussion with literature data, even regarding left MIDCAB?
  2. Thank you for this comment. One of our adverse events – a revascularization after surgery – was presented in the discussion. (page 10; line 279 – 281). We added our adverse events as suggested in the discussion (page 10, line 282 – 287) and added literature data as well.
  3. See page 10, line 282 – 287

Round 2

Reviewer 3 Report

Thank you for the answers. 

I think that 

  1. Thank you for your comment. The expected rate of adverse peri-procedural events and the 24-months events would probably be comparable to the presented data of Davierwala et al (DOI: 10.1016/j.jtcvs.2020.12.149) who provided a 22 years experience in over 2000 patients receiving MIDCAB surgery. Our provided sample size may be under-powered but is nevertheless very interesting for inspirational reasons to other groups to evaluate their results in this fairly rare surgical method.

merits a note in the methods.

Best regards